# The Effects of Carbohydrate Mouth Rinse on Psychophysiological Responses and Kinematic Profiles in Intermittent and Continuous Small-Sided Games in Adolescent Soccer Players: A Randomized, Double-Blinded, Placebo-Controlled, and Crossover Trial

**DOI:** 10.3390/nu16223910

**Published:** 2024-11-15

**Authors:** Yusuf Soylu, Paweł Chmura, Ersan Arslan, Bulent Kilit

**Affiliations:** 1Faculty of Sport Sciences, Tokat Gaziosmanpasa University, Tokat 60250, Türkiye; ersan.arslan@gop.edu.tr (E.A.); bulent.kilit@gop.edu.tr (B.K.); 2Department of Team Games, Wroclaw University of Health and Sport Sciences, 51-612 Wroclaw, Poland; pawel.chmura@awf.wroc.pl

**Keywords:** game-based training, physical enjoyment, perceived exertion, interval training, soccer performance, nutritional supports

## Abstract

Background: Mouth rinsing (MR) with a carbohydrate solution is one of the most popular methods athletes use to improve their game-based performance due to its acute ergogenic effect. Objectives: This study aimed to evaluate the effects of the carbohydrate MR intervention on psychophysiological responses and kinematic profiles during intermittent (_INT_) and continuous (_CON_) 4-a-side small-sided soccer games (SSGs). Methods: Thirty-two adolescent soccer players (age: 16.5 ± 0.5 years) played six bouts of 4-a-side SSGs with MR_INT_ or MR_CON_ at 3-day intervals in a randomized, double-blinded, placebo-controlled, and crossover study design. Psychophysiological responses and kinematic profiles were continuously recorded during all games. The rating of perceived exertion (RPE), the rating scale of mental effort (RSME), and the physical enjoyment scores (PES) were also determined at the end of each game. Results: The MR_CON_ induced higher psychophysiological responses such as RPE, internal training load (ITL), and RSME (*p* ≤ 0.05, *d* values ranging from 0.50 to 1.04 [small to moderate effect]). Conversely, the MR_INT_ induced higher PES (*p* ≤ 0.05, *d* values = 1.44 [large effect]) compared to MR_CON_. Although the MR intervention led to similar improvements in the performance of 4-a-side MR_INT_ and MR_CON_, there was no significant difference between the groups. Conclusions: Our results suggest that the MR intervention can be used as an effective ergogenic supplement for acute game performance enhancement, regardless of the game’s structure.

## 1. Introduction

Soccer, a physically demanding sport, requires players to sustain high proficiency levels during matches. Nutritional strategies to enhance soccer performance, primarily through carbohydrate (CHO) interventions, are crucial for players to optimize their energy, recovery, and physical capabilities [1]. MR with CHO and CHO ingestion are two strategies coaches and practitioners use to enhance players’ game performance [2,3]. In contrast to CHO ingestion, MR provides a practical alternative that offers performance benefits without digestion, as the CHO solution is swished into the oral cavity and subsequently expectorated [4]. Therefore, MR may effectively enhance soccer performance, which depends on factors such as the type of exercise, the player’s nutritional status, and the concentration of the CHO solution.

Small-sided soccer games (SSGs) are a standard training method in soccer, offering a way to enhance players’ psychophysiological, technical, and tactical skills [5]. The SSGs primarily design continuous games (_CON_) without repetitions or rest intervals, and intermittent games (_INT_) with repetitions and rest intervals in two main formats, which modify the number of players, pitch dimensions, and duration to simulate match scenarios [6]. Previous studies have indicated that _CON_ and _INT_ training provide no significant advantage, thus supporting soccer-specific training [7,8]. Both methodologies have demonstrated efficacy in improving the bio-motor abilities of young soccer players by producing different physiological responses [9]. Recent studies have focused on alternative tools that enhance player productivity, including coaching behaviour [10], cognitive training [11], and nutritional support [1]. However, nutritional strategies, including CHO, are essential for enhancing soccer performance as potential ergogenic aids, particularly in high-intensity sports, such as soccer [12,13]. Integrating nutritional strategies such as MR within SSGs to enhance performance metrics further has yet to be thoroughly investigated, representing an unaddressed need for optimizing soccer training methodologies.

MR with CHO, one of the most popular methods recently used to improve soccer performance, is a more practical, low-cost, and effective CHO consumption method compared to other CHO intake forms, such as solid, liquid, and gel [14]. CHO methods include the MR technique [15], which involves briefly swishing a CHO solution in the mouth before spitting it out without ingestion to enhance soccer performance [16,17]. However, Hartley [4] stated that MR was used at different rates (6–18%), and the 6–6.5% concentration range was significantly improved during exercise. Previous studies [14,18] have explained that the underlying mechanism of MR is the activation of neural pathways in the brain during oral activity, potentially reducing perceived effort and enhancing exercise performance. Furthermore, recent systematic reviews have examined the practical performance benefits of MR and the absence of gastrointestinal discomfort, which are typical of different CHO intake methods [4,19]. Therefore, the expeditious and productive utilization of MR in complex sports like soccer may contribute to developing an appropriate training strategy.

For a long time, coaches have utilised SSGs to develop high-soccer game demands such as game skills, technical–tactical awareness, and player interactions during matches [20]. Despite the limited studies on soccer players’ physical performance, the ergogenic effects of MR on SSG performance still need to be studied in more detail. Previous studies demonstrated that MR improves soccer performance regarding mean power, endurance, and sprinting [21,22,23]. Contrary to these recent studies, Bortolotti et al. [24] reported that the sprint performance of soccer players did not improve after the MR intervention. Nevertheless, Rollo et al. [3] observed that MR notably enhanced the self-selected jogging pace and exhibited an 86% probability of improving performance in repeated sprints compared to a placebo group. Additionally, a recent study concluded that single and serial MR did not enhance the distance of the Yo-Yo Intermittent Recovery Test Level 1 [25]. A literature review indicated that rinsing the oral cavity with CHO elicits a distinct metabolic response compared to CHO ingestion [14]. Furthermore, another recent systematic review stated that MR with CHO improves physical performance through psychophysiological effects, directly linking better task-specific activity to activation of the primary sensorimotor cortex due to oral CHO transmission [1]. Therefore, clarifying the specific effects of MR in various SSG formats may significantly improve players’ performance during training. As noted by Rollo et al. [3], while MR demonstrates efficacy in enhancing performance during intermittent exercises, its influence on soccer-specific formats, namely _INT_ and _CON_ SSGs, requires further investigation, thus highlighting the pioneering nature of this investigation. Finally, current studies show that more studies are needed on the effect of the MR mechanism on soccer performance. However, no study has been conducted on the impact of SSG performance, such as _INT_ and _CON_ SSGs. Therefore, this study aimed to evaluate the effects of the MR intervention on psychophysiological responses and kinematic profiles during _INT_ and _CON_ methods of SSGs.

## 2. Materials and Methods

### 2.1. Subjects

Before starting the study, the sample size of the study was estimated using G*Power software (G-Power, version 3.1.9.7, University of Düsseldorf, Düsseldorf, Germany) with a partial effect size of 0.30, power of 0.8, and correlation of 0.5 with a *p*-value of 0.05. This software demonstrated that the minimum sample size required for the study was 24 participants. Therefore, thirty-two male well-trained soccer players (mean ± SD age: 16.5 ± 0.5 years, height: 173.6 ± 6.0 cm, weight: 58.9 ± 7.2 kg, training age: 4.9 ± 0.4 years) randomly played six bouts of 4-a-side SSGs with MR_INT_ or MR_CON_ formats at 72 h intervals to avoid the interventions’ potential negative or positive effects on game-based performance (Figure 1). The inclusion criteria were as follows: (i) players had been involved in soccer training (4 training units [≈90 min] a match per week) for more than three years; (ii) players were not injured during or in the month before the study; and (iii) players did not take any drug or ergogenic support during the study. Before signing the informed consent form, the players and their parents were informed of the safety, risks, and measurement procedures. However, they were not fully informed of the study’s true purpose to avoid influencing the study results. All players stated that they had not previously practiced MR and were unaware of its benefits. All the participants and their parents provided written informed consent for participation. This study was approved by the Research Ethics Committee (protocol code: 11.06.2024-440703) and was conducted according to the Declaration of Helsinki.

### 2.2. Study Design

In the present study, a randomized, double-blinded, placebo-controlled, and crossover study design was used to assess the effects of MR intervention on the psychophysiological responses and kinematic profiles of 4-a-side SSG with MR_INT_ or MR_CON_ formats in adolescent soccer players. Before the MR intervention, the players completed the Yo-Yo Intermittent Recovery Test Level 1 (YYIRT-1) to divide them into two groups according to their aerobic fitness levels to avoid having unbalanced 4-a-side SSG groups in both games. Following a 15 min standardized warm-up section, consisting of jogging, sprinting, and integrating soccer-specific actions, the players performed 4-a-side SSG with MR_INT_ or MR_CON_ formats on a standard artificial grass pitch (Figure 2). After a 15 min standardized warm-up section, young players were instructed to rinse the solution (10 s with 25 mL of either a 6.4% maltodextrin solution (MR) or an identically flavoured aspartame solution (PLA)) in their mouth for 10 s and then expectorate back into the cup. The order of different SSG (MR_INT_ or MR_CON_) and solutions (MR or PLA) was determined by randomization (www.randomization.com). Each game was then randomly played with six bouts of 4-a-side SSG, either MR_INT_ or MR_CON_ formats, at 72 h intervals to provide a psychophysiological recovery effect to the players. MR_INT_ and MR_CON_ were performed on standard artificial grass pitch under similar weather conditions (28–30 °C, temperature and 33–35%, humidity). Physiological responses and kinematic profiles were continuously monitored and recorded during all SSG. Psychological responses, such as RPE, RSME, and PES, were also determined at the end of each SSG. The players were given no tactical instructions or specific rules throughout the game, and numerous balls were placed around the pitch to ensure a continuous game. Their coaches verbally encouraged maximal effort during all SSG [10].

### 2.3. Procedures

#### 2.3.1. Anthropometric Measurements

Before breakfast, players’ weight and height were measured using a body composition analyser (BC-418MA, Tanita Corp., Tokyo, Japan). Using the bioelectrical impedance method, this analyser uses multiple frequencies (ranging from 1 to 50 kHz) to measure detailed body composition measurements.

#### 2.3.2. Yo-Yo Intermittent Recovery Level 1 Test

The Yo-Yo Intermittent Recovery Test Level 1 (YYIRTL-1), an acoustically and reliably progressive test, was used to evaluate aerobic capacity [26] using the procedure recommended by Bangsbo et al. [27]. A heart rate (HR) monitor (Polar V800 Polar OY, Kempele, Finland) was used to measure each player’s HR throughout the test. The highest HR response was recorded as the maximum HR value in the YYIRTL-1 test. After the test, the estimated maximal oxygen consumption (VO_2max_) was calculated using the following formula [27].
Estimated VO_2max_ = 36.4 + (0.0084 × covered distance in YYIRT-1)

#### 2.3.3. Psychophysiological Responses

Validated and reliable psychophysiological scales, the cheapest and most practical method, have been recently used in sports psychology to evaluate athletes’ psychological states before, during, and after physical performance [28,29]. The rating of perceived exertion (RPE), frequently used as an indicator of exercise intensity in both _INT_ and _CON_ formats of SSGs, was determined using a category ratio scale (CR-10) to calculate the internal training load (ITL) immediately after the completion of each SSG session. The session RPE (sRPE) quantifies the ITL by multiplying the entire training sRPE using the CR-10 scale by its duration [30]. The rating scale mental effort (RSME), which has a good relationship with performance, is a single item that assesses mental workload ranging from 0 (no effort) to 150 (extreme effort) [31,32]. Furthermore, this scale has been recently used in sports psychology to evaluate effort in the relationship between emotions and performance during competition [31]. All players completed the physical enjoyment scale (PES). The scale includes eight items scored on a 1–7 Likert scale and has been validated as a marker of the level of enjoyment in Turkish adolescents and adults [33]. The players individually filled out psychophysiological scales, such as RPE, RSME, and PES, immediately after each training session to avoid being influenced by their teammates’ scores.

#### 2.3.4. Kinematic Profiles

The physiological responses and time-motion characteristics of the players were monitored using a portable Global Navigation Satellite System (GNSS) with a frequency of 10 Hz and a triaxial accelerometer with a frequency of 100 Hz (STATSports, Apex, Londonderry, Northern Ireland). GNSS Apex units with dimensions of 30 mm (wide) × 80 mm (high) and weighing 48 g were placed on the back of the participant midway between the scapulas. These units have inter-unit reliability ranging from excellent to suitable for monitoring short-distance intermittent running activities [34]. GNSS data recorded by the Apex units were downloaded and analysed by the STATSports Apex Software (Apex 10 Hz, Sonra v4.2.1). Six speed zones—standing (0–0.7 km·h^−1^), walking (0.7–7.2 km·h^−1^), jogging (7.2–14.4 km·h^−1^), running (14.4–19.8 km·h^−1^), high-speed running (19.8–25.1 km·h^−1^), and sprinting (>25.1 km·h^−1^)—and the other variables related to time-motion characteristics were selected for kinematic data analysis in line with previous study [35].

#### 2.3.5. Supplementation Procedures

A double-blinded design was used in this study. Except for the supervisor investigator, the other researchers were blinded to prepare a 6.4% maltodextrin (Protein Ocean, Ankara, Türkiye) solution (MR) or an identically flavoured solution (PLA). The PLA solution contained pure water. Both solutions, kept at room temperature and prepared daily, were made indistinguishable by incorporating a non-calorific artificial sweetener, sucralose, which the supervisor investigator added in equal amounts. A 25 mL bolus of 6.4% maltodextrin solution was utilized in a pre-weighed plastic cup, with sucralose water tested as PLA for each rinse solution. Before expectorating the solution back into the cup, the players were instructed to swirl the solution throughout their oral cavity for ten seconds. The solution was measured before and after each rinse using an electronic balance to assess the potential quantity of the solution that could have remained in the oral cavity. Players were not informed about the effects of the MR intervention and the study’s primary purpose until the completion of the study.

### 2.4. Statistical Analyses

All data are presented as the mean ± standard deviation. Data distribution was assessed using the Shapiro–Wilk test. A paired *t*-test was used to compare differences in psychophysiological responses regarding RPE, PES, RSME, and ITL between the MR_INT_ and MR_CON_ conditions. Repeated measures analysis of variance (ANOVA) was used to compare interactions and main effects on psychophysiological responses and kinematic profiles. The effect sizes (Cohen’s *d*) were also calculated for each dependent variable. Cohen’s *d* values were considered trivial (<0.20), small (0.20–0.59), moderate (0.6–1.19), large (1.2–1.99), and very large (≥2.0) [36]. The 95% Confidence Interval (95% CI) was calculated as the difference between the mean values of the measured variables. SPSS version 24.0 was used to conduct all statistical analyses (SPSS, version 24.0; SPSS Inc., Chicago, IL, USA). Statistical significance was set at *p* ≤ 0.05.

## 3. Results

Figure 3 shows that MR_CON_ induced higher psychophysiological responses such as RPE (t = −2.460; *p* = 0.020; %95 CI = −1.06–−0.09, *d* = 0.50 [small effect]), ITL (t = −2.458; *p* = 0.020; %95 CI = −25.39–−2.36, *d* = 0.50 [small effect]), and RSME (t = −5.614; *p* = 0.000; %95 CI = −27.41–−12.80, *d* = 1.03 [moderate effect]) than MR_INT_. Conversely, the MR_INT_ induced higher PES (t = −5.812; *p* = 0.000; %95 CI = 5.55–11.56, *d* = 1.44 [large effect]) than MR_CON_.

Table 1 demonstrates that significant changes were observed in the measured physiological responses and kinematic profiles (*p* ≤ 0.05, *d* values ranging from trivial to moderate effects) in terms of within-group comparisons. The between-group comparison also demonstrated that the impact of MR in the SSG_INT_ and SSG_CON_ conditions was similar in terms of physiological responses and kinematic profiles (*p* ≥ 0.05).

## 4. Discussion

This study aimed to evaluate the effects of the MR intervention on psychophysiological responses and kinematic profiles during _INT_ and _CON_ 4-a-side SSGs. The strength of this study is that it attempts to determine whether the game structure can differentiate players’ results. The applied strategy was assumed to improve game results owing to its acute ergogenic effect. Our main finding was that the MR intervention led to similar improvements in the performance of both _INT_ and _CON_ methods of 4-a-side SSGs; however, there was no significant difference between the groups.

In the current study findings of the MR_CON_, we recorded that this game structure induced higher psychophysiological responses such as RPE, ITL and RSME. Farhani et al. [37] emphasized that in the design of SSG-based training programs, trainers must incorporate extended uninterrupted sessions to enhance physical exertion and the training load substantially. Moreover, considering the limitations of game-based training in soccer, previous studies have assessed the impact of MR on RPE during intermittent high-intensity exercise [3,17,21]. When using similar protocols and populations, these studies found no effect of a single MR on the high-intensity performance of soccer players (YYIRTL-1). Given the lack of MR and the lack of consideration of its impact on continuous exercises in soccer, in the literature, most of the protocols implemented to date using the described strategy have been in studies of endurance sports such as cycling, triathlon, and others. Luden et al. [38] reported that the MR improved performance in high-intensity trials following extended cycling periods. On the other hand, among trained cyclists and triathletes in a CHO-depleted state, MR intervention did not affect perceptual responses during high-intensity exercise [39]. In turn, Brietzke et al. [40] demonstrated that the MR could mitigate the effects of mental fatigue and potentially enhance overall maximal incremental test performance without affecting cortical functions in well-trained male cyclists. However, Pires et al. [41] revealed that the MR did not reduce central fatigue, decrease perceived effort, and enhance performance during progressive exercise in recreational cyclists. A recent systematic review [42] suggested that the improvements in MR performance may be due to central mechanisms rather than being attributed to glycogen stores and the availability of external CHO during early exercise hours. Moreover, MR with CHO does not directly supply energy. However, it can influence the central nervous system to signal muscles to work effectively for extended periods, thereby delaying the perception of physical fatigue and maintaining muscle productivity [19]. In conclusion, _CON_ games, as opposed to _INT_ games, require physical and mental planning that demands sustained effort for prolonged endurance, so that central fatigue-regulating systems can operate more actively. Hence, with this in mind, further research is needed that could confirm or deny the positive impact of MR on psychophysiological responses such as RPE, ITL, and RSME.

Moving on to games performed in an intermittent structure, conversely, the SSG_INT_ induced higher PES compared to MR_CON_. Here, we recorded the largest effect between the MR and PLA groups. To our knowledge, no study has examined the acute effects of MR_CON_ on PES; only one study has investigated pleasure responses alone during the intermittent type of exercises [3]. This study also demonstrated that the Loughborough soccer test showed that the enjoyment received during the Intermittent Shuttle Test was not significantly affected by MR intervention. Furthermore, MR did not alter the perception of pleasure in endurance exercises conducted in diverse groups of athletes, including trained male cyclists and recreational runners [39,40,41,42,43]. Rollo et al. [44] postulated that pleasure or perceived sensation activation might manifest as changing exercise behaviour, such as increased running speed. In addition, an alternative hypothesis suggests that the MR towards the conclusion of prolonged intermittent running could alter the equilibrium between excitation and inhibition of centrally mediated motor output, enhancing running performance [45]. Previous studies [46,47] have clarified that enhanced perceived enjoyment may be derived from participant satisfaction, which is essential for self-determined motivation during MR_INT_. This confirms that the strategy yields good results and should be included in the training process. This is especially important in team sports such as soccer, basketball and handball, where physical effort is mixed and intermittent.

We also observed in the MR_INT_ group a moderate positive effect of MR on performance in variables such as total distance, standing, sprinting, and relative metabolic load. This is consistent with previous studies that reported that routine MR increases the self-selected submaximal running speed [3,44]. To give an illustration, while a study such as that of Rollo et al. [3] has demonstrated MR’s potential in improving running performance, the specific impact of MR on soccer’s particular kinematic demands, especially within a variety of formats of SSGs, has yet to be determined. Previous studies have demonstrated that MR indicates that the interaction between CHO and the oral mucosa elicits a response from receptors associated with the central nervous and reward systems [48], subsequently enhancing performance [15]. However, some studies concluded that MR with CHO had no significant effect on the distance covered by recreational runners [43,49]. Furthermore, it is somewhat surprising that in the MR_CON_ group, lower values in the number and length of sprints were recorded than in the PLA group, with moderate effects. This needs to be clarified in future studies. One of the variables that could have influenced this finding was that our intervention was performed under hot conditions. Although our results showed a positive effect of using MR, there is some evidence that any benefit of MR is lost when exercise is performed in a warm environment [50,51]. Further research is required to clarify this condition.

### 4.1. Practical Applications

This study revealed that the CHO solution with MR has shown different effects on psychophysiological and kinematic responses during different SSG formats, such as _CON_ and _INT_. This study suggested that MR with CHO enables players to cope with fatigue during prolonged games. Additionally, _INT_ games reduce prolonged physical and cognitive exertion, enhancing players’ enjoyment. Moreover, the intrinsic nature of _INT_ game mitigates prolonged physical and mental exertion periods, enhancing the participants’ enjoyment of the activity. Given the psychophysiological and kinematic differences between _INT_ and _CON_ games, coaches can determine the relevance and inclusion of MR strategies in their ergogenic protocols to meet specific training goals or player needs. Therefore, practitioners, and strength and conditioning coaches considering using MR with CHO strategies to improve player performance should carefully evaluate the effect of MR with CHO solutions in MR_INT_ and MR_CON_ games.

### 4.2. Limitations

The authors fully know the limitations and various variables that may have influenced the analysis results. The current study only assessed young, well-trained male soccer players still developing biological and psychomotor capabilities, limiting its applicability to competitive soccer players from different divisions to varying fitness levels, female soccer players, or older players. Although this study only used a 4-a-side SSG format, the researchers could investigate whether the game influences the performance of MR with CHO using several variables, such as the number of players, rules, court dimensions, and coach encouragement, which are known to influence SSG performance. Finally, environmental temperatures can significantly affect physiological and psychophysiological responses during exercise, as the players are trained in hot conditions. Moreover, this study measured HR, PES, and RPE responses but could also consider psychophysiological variables such as blood lactate level, motivation, cognition, and anxiety, which MR might impact and affect performance. Accepting this limitation and recommending future research to investigate the effects of CHO with MR in a significantly broader target demographic group would increase our understanding of the ergogenic effects of MR. In addition, it would be worth using MR to examine the influence of this nutritional strategy on breaking players’ fatigue barrier and shifting the psychomotor fatigue threshold towards higher physical loads, at which the fastest reaction, anticipation, and decision-making occur.

## 5. Conclusions

Our results support that regardless of the game’s structure, the MR intervention can be used as an effective ergogenic supplement for acute game performance enhancement in young soccer players. The _CON_ game elicited higher psychophysiological responses, including RPE, ITL, and RSME, whereas the _INT_ game elicited greater PES. Although the MR intervention similarly enhanced performance in the 4-a-side _INT_ and _CON_ games, no significant differences were observed between the groups. Clarifying these responses could offer a complete understanding of the influence of MR on soccer performance. However, this study focused on the acute responses to MR during the micro-cycle. From a practical perspective, future research should explore the different dose effects of regular MR use throughout the training adaptations and recovery.

## Figures and Tables

**Figure 1 nutrients-16-03910-f001:**
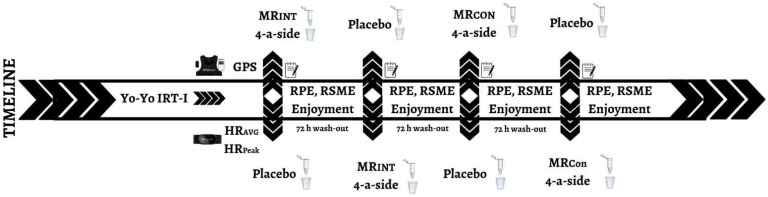
Study design.

**Figure 2 nutrients-16-03910-f002:**
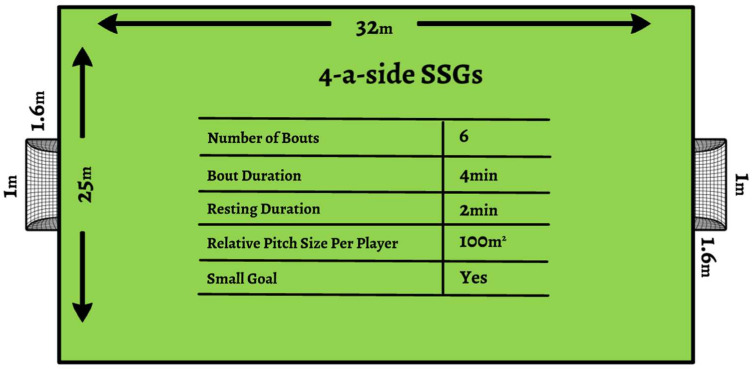
Structure of SSGs.

**Figure 3 nutrients-16-03910-f003:**
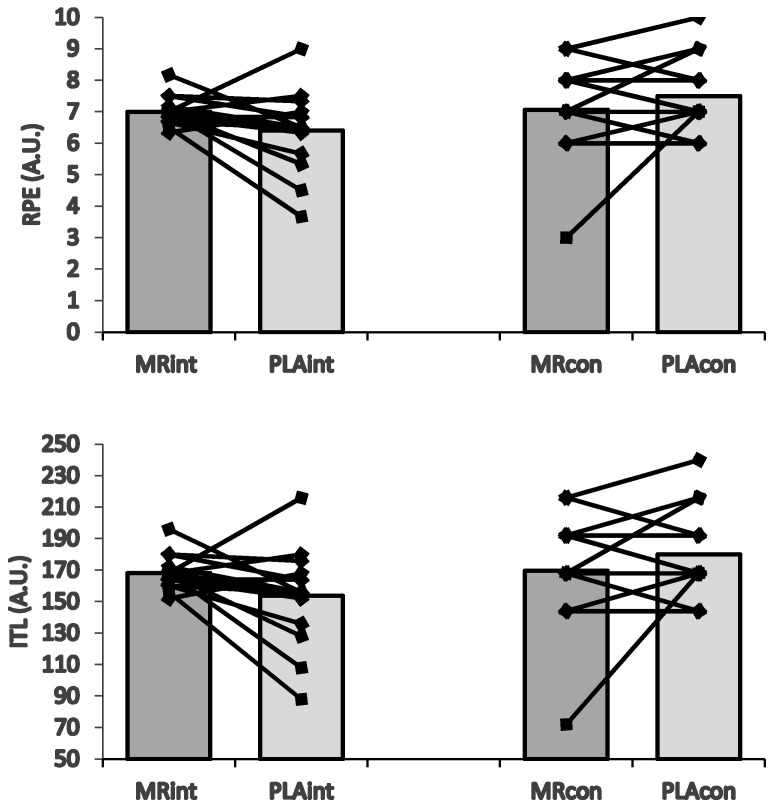
The MR intervention on psychophysiological responses during _INT_ and _CON_ 4-a-side SSGs.

**Table 1 nutrients-16-03910-t001:** Effect of MR on performance responses of the players during 4-a-side SSG_INT_ or SSG_CON_.

	SSG_INT_				SSG_CON_			
	MR	PLC	%95 CI	*d*	MR	PLC	%95 CI	*d*
Peak Heart Rate (beat·min^−1^)	186.1 ± 7.7	186.6 ± 6.8	−12.10–−0.35	0.07	191.7 ± 8.8	189.6 ± 10.2	−10.81–8.58	0.22
Average Heart Rate (beat·min^−1^)	174.4 ± 9.2	172.6 ± 6.5	−9.84–6.84	0.22	174.9 ± 13.6	173.1 ± 7.8	−6.68–21.71	0.16
Total Distance (m)	1985.5 ± 133.9 *	1864.4 ± 131.1	−673.88–−400.81	0.91	2522.8 ± 172.3 *	2578.9 ± 183.0	−854.59–574.33	0.31
Standing (m)	783.1 ± 96.5 *	707.1 ± 115.5	−187.91–22.56	0.71	865.8 ± 140.2 *	891.0 ± 145.7	−297.22–70.61	0.17
Walking (m)	189.2 ± 36.0 *	177.2 ± 35.3	−66.66–−9.96	0.34	217.6 ± 51.2 *	214.9 ± 47.9	−70.56–5.00	0.05
Jogging (m)	169.2 ± 38.5 *	164.6 ± 43.2	−64.66–−9.81	0.11	196.6 ± 54.9 *	197.9 ± 50.7	−72.07–5.40	0.02
Running (m)	64.5 ± 25.7	66.9 ± 28.2	−30.72–6.26	0.09	76.7 ± 28.9	78.4 ± 26.6	−32.67–9.69	0.06
High-Speed Running (m)	18.2 ± 14.7	20.2 ± 17.8	−9.87–8.16	0.12	19.0 ± 19.1	19.9 ± 15.3	−14.90–15.49	0.05
Sprinting (m)	0.3 ± 0.7 *	1.4 ± 2.4	−0.05–0.76	0.62	0.0 ± 0.0	0.0 ± 0.0	0.16–2.71	0.00
Peak Speed (km·h^−1^)	20.2 ± 1.1	20.4 ± 1.3	−2.45–−0.82	0.20	21.8 ± 1.3	21.7 ± 1.3	−2.45–0.04	0.11
Average Speed (km·h^−1^)	10.9 ± 0.3	11.1 ± 0.4	−0.25–0.24	0.25	10.9 ± 0.4	10.9 ± 0.3	−0.18–0.34	0.03
Number of Sprints (n)	48.8 ± 10.0 *	44.3 ± 6.9	−9.15–6.41	0.52	50.2 ± 11.1 *	58.1 ± 12.4	−21.77–5.85	0.67
Length of Sprint (m)	430.3 ± 89.0 *	408.6 ± 84.9	−117.98–41.99	0.25	468.3 ± 102.4 *	519.7 ± 11.8	−187.83–34.29	0.70
Relative Metabolic Load	10.9 ± 0.9 *	10.2 ± 0.8	−3.49–1.84	0.83	13.6 ± 1.0 *	14.0 ± 1.1	−4.65–2.89	0.36
Metabolic Power Average	42.7 ± 3.3	42.1 ± 3.4	31.38–35.25	0.17	9.4 ± 0.7	9.6 ± 0.7	30.57–34.59	0.22
High-Metabolic Power Distance (m)	521.1 ± 91.9 *	490.7 ± 88.8	−144.77–21.85	0.34	582.6 ± 105.7 *	603.0 ± 107.3	−193.11–31.38	0.19

Data are mean ± SD. * Significant difference from PLC. *p* ≤ 0.05.

## Data Availability

The data that support the findings of this study are available from the corresponding author upon reasonable request.

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
