# Peer review of "The Effects of Carbohydrate Mouth Rinse on Psychophysiological Responses and Kinematic Profiles in Intermittent and Continuous Small-Sided Games in Adolescent Soccer Players: A Randomized, Double-Blinded, Placebo-Controlled, and Crossover Trial"

_nutrients, 2024, doi:10.3390/nu16223910_

Round 1
Reviewer 1 Report
Comments and Suggestions for Authors
Denote somewhere in the title that these are adolescent athletes.
Since “SSG” is in both abbreviations, it is not needed as it hampers readability of the manuscript. Change to “INT” and “CON”, the reader will know it is a soccer game.
Single blinded?
Line 44: I do not think authors need to spend so much time discussing the SSG in the introduction as it is not really your intervention.
Authors have cited a lot of research looking at the acute explosive effects of CHOmr (e.g. sprinting). What is the rationale for why CHOmr would help over the course of an entire game when the effect is lost so quickly?
Authors should remove discussion about energy and metabolism in the context of CHOs for this, as CHOmr is a different neural pathway which authors have highlighted.
Did you get informed assent from the parents? Consenting a minor on their own is not customary.
Figure 1- You cannot read the text, even at 150% magnification. I suggest using abbreviations in the figure so the text can be larger. Then, you can spell out the abbreviations in the caption. The figure looks quite nice though!
“or crossover study design?”. What is crossover?
Line 119: How are they identically flavored? Was aspartame included in the maltodextrin solution as well?
“A single-blinded design was used in this study. Except for the supervisor investiga- 171
tor, the other researchers were blinded to prepare a 6.4% maltodextrin (Protein Ocean, 172
Türkiye) solution (CHOMR) or an identically flavoured aspartame solution (PLA).” Were researchers blinded and not participants? This seems double-blinded based on this statement.
Line 187: making a figure showing the structure of the games would be helpful
What post-hoc test was used?
Figure 2. These figure look fantastic. I might suggest making the large mean bars more transparent so you can see the individual lines better.
I think an argument author should make in the discussion is that CHOmr may be more beneficial in a fatigued state.
Line 283: authors only saw moderate effects in sprint number and distance. The rest were small. Please remove these from the discussion as a “moderate” effect if the effect size is not >0.5.
Line 294: CHO ingestion should not be discussed in this manuscript as they did not ingest them. Glycogen is not within the mechanism for how CHOmr works.
Author should spend more time explaining why the limitations matter.
Why discuss outcomes that were not measured in the conclusion?
Author Response
We are very grateful for your patience and the opportunity to revise the paper. We would like to respond to your overall evaluation before revising the article point by point based on your comments.
Regards
Reviewer: Denote somewhere in the title that these are adolescent athletes.
Author: Added.
Reviewer: Since “SSG” is in both abbreviations, it is not needed as it hampers readability of the manuscript. Change to “INT” and “CON”, the reader will know it is a soccer game.
Author: Changed throughout the manuscript.
Reviewer: Single blinded?
Author: Changed as “double-blind”
Reviewer: Line 44: I do not think authors need to spend so much time discussing the SSG in the introduction as it is not really your intervention.
Author: The sentence was removed.
Reviewer: Authors have cited a lot of research looking at the acute explosive effects of CHOmr (e.g. sprinting). What is the rationale for why CHOmr would help over the course of an entire game when the effect is lost so quickly?
Author: Detailed explained into the manuscript.
Reviewer: Authors should remove discussion about energy and metabolism in the context of CHOs for this, as CHOmr is a different neural pathway which authors have highlighted.
Author: We are totally agree with you, the sentence was removed.
Reviewer: Did you get informed assent from the parents? Consenting a minor on their own is not customary.
Author: Thank you very much for your patient. Added.
Reviewer: Figure 1- You cannot read the text, even at 150% magnification. I suggest using abbreviations in the figure so the text can be larger. Then, you can spell out the abbreviations in the caption. The figure looks quite nice though!
Author: Thank you very much reconfigured.
Reviewer: “or crossover study design?”. What is crossover?
Author: In a crossover study design, two interventions (e.g., exercise, methods, and training) are provided to subjects at different time periods, and the sequence of intervention is randomized for each subject. Similarly, the research design of the current study includes two different groups and two different interventions.
Reviewer: Line 119: How are they identically flavored? Was aspartame included in the maltodextrin solution as well?
Author: Both formulations were designed to possess similar flavor profiles to minimize organoleptic differences between the solutions. However, no aspartame or comparable artificial sweeteners were incorporated, and a maltodextrin solution was prepared to render its inherent flavor indistinguishable from that of the placebo solution.
Reviewer: “A single-blinded design was used in this study. Except for the supervisor investigator, the other researchers were blinded to prepare a 6.4% maltodextrin (Protein Ocean, Türkiye) solution (CHOMR) or an identically flavoured aspartame solution (PLA).” Were researchers blinded and not participants? This seems double-blinded based on this statement.
Author: Thank you, we changed.
Reviewer: Line 187: making a figure showing the structure of the games would be helpful
Author: Thank you very much for your suggestion, we added a figure. We hope you like it.
Reviewer: What post-hoc test was used?
Author: We do not use any post-hoc test. Instead of post-hoc test, we used effect size due to we have two groups.
Reviewer: Figure 2. These figure look fantastic. I might suggest making the large mean bars more transparent so you can see the individual lines better.
Author: Thank you very much. Changed.
Reviewer: I think an argument author should make in the discussion is that CHOmr may be more beneficial in a fatigued state.
Author: Thank you very much again. That’s fantastic insight. We added.
Reviewer: Line 283: authors only saw moderate effects in sprint number and distance. The rest were small. Please remove these from the discussion as a “moderate” effect if the effect size is not >0.5.
Author: Thank you very much. We especially focused on kinematic responses such as total distance (0.91), number of sprints (0.52) and relative metabolic load (0.83). We have already use the effect size >0.5.
Reviewer: Line 294: CHO ingestion should not be discussed in this manuscript as they did not ingest them. Glycogen is not within the mechanism for how CHOmr works.
Author: We are totally agree with you, the sentence was changed.
Reviewer: Author should spend more time explaining why the limitations matter.
Author: Reorganized our limitations
Reviewer: Why discuss outcomes that were not measured in the conclusion?
Author: Reorganized our conclusion.

Reviewer 2 Report
Comments and Suggestions for Authors
Soylu et al. provide a study aimed to evaluate the effects of the CHOMR intervention on psychophysiological responses and kinematic profiles during intermittent (INT) and continuous (CON) 4-a-side small-sided 15 soccer games (SSG). I consider that it could be a relevant research for training and nutrition in this discipline and could provide great practical contributions. Next, I will point out some points for improvement:
· Abstract: it would be appreciated if each section could be announced with its title background/objectives/methodology/results.
· Introduction: I am of the opinion that the introduction is well-presented and comprehensible, as it delineates the issues that the study will address. Nevertheless, I am of the opinion that the final paragraph should reiterate the study's objective.
· Material/Methods: The authors present a concrete intervention protocol with carefully formatted and written subsections, and most of the references are very up to date, which means that the study may have practical interventions in the future. However, I consider that the psychological variables evaluated should be explained in greater depth.Also, I wonder if it would be possible to indicate the level of professionalization of the subjects, it would be necessary and would improve the description of the participants' point (2.1)
· Results: I am not required to make any modifications to the tables and graphs, as they are well-written and comprehensible.
· Discussion: It is highly concrete and is compared to numerous studies; therefore, I regard it as a commendable work. Nevertheless, I am curious as to whether they have considered incorporating a point before the conclusions that more explicitly delineates the practical applications.
I believe that they have done a good job and have approached the study well. I would improve those aspects simply for a better understanding of what is aimed at.
Author Response
We are very grateful for your patience and the opportunity to revise the paper. We would like to respond to your overall evaluation before revising the article point by point based on your comments.
Regards
Reviewer: Soylu et al. provide a study aimed to evaluate the effects of the CHOMR intervention on psychophysiological responses and kinematic profiles during intermittent (INT) and continuous (CON) 4-a-side small-sided 15 soccer games (SSG). I consider that it could be a relevant research for training and nutrition in this discipline and could provide great practical contributions. Next, I will point out some points for improvement:
Author: Thank you very much for your valuable suggestions.
Reviewer: Abstract: it would be appreciated if each section could be announced with its title background/objectives/methodology/results.
Author: Reorganized.
Reviewer: Introduction: I am of the opinion that the introduction is well-presented and comprehensible, as it delineates the issues that the study will address. Nevertheless, I am of the opinion that the final paragraph should reiterate the study's objective.
Author: Thank you very much for your patient. Added.
Reviewer: Material/Methods: The authors present a concrete intervention protocol with carefully formatted and written subsections, and most of the references are very up to date, which means that the study may have practical interventions in the future. However, I consider that the psychological variables evaluated should be explained in greater depth.Also, I wonder if it would be possible to indicate the level of professionalization of the subjects, it would be necessary and would improve the description of the participants' point (2.1)
Author: Thank you very much for your patient. Added.
Reviewer: Results: I am not required to make any modifications to the tables and graphs, as they are well-written and comprehensible.
Author: Thank you very much.
Reviewer: Discussion: It is highly concrete and is compared to numerous studies; therefore, I regard it as a commendable work. Nevertheless, I am curious as to whether they have considered incorporating a point before the conclusions that more explicitly delineates the practical applications.
Author: Thank you very much. We added.
Reviewer: I believe that they have done a good job and have approached the study well. I would improve those aspects simply for a better understanding of what is aimed at.
Author: Thank you very much.

Round 2
Reviewer 1 Report
Comments and Suggestions for Authors
My comments have been addressed.
Author Response
We are very grateful for your patience and the opportunity to revise the paper. We would like to respond to your overall evaluation before revising the article point by point based on your comments.
Reviewer: Denote somewhere in the title that these are adolescent athletes.
Author: Added. Players are defined as adolescents in the title and text.
Title: The Effects of Carbohydrate Mouth Rinse on Psychophysiological Responses and Kinematic Profiles in Intermittent and Continuous Small-sided Soccer Games in Adolescent Soccer Players: A Randomized, Double-Blinded, Placebo-Controlled, and Crossover Trial
Reviewer: Since “SSG” is in both abbreviations, it is not needed as it hampers readability of the manuscript. Change to “INT” and “CON”, the reader will know it is a soccer game.
Author: Changed throughout the manuscript.
Instead of SSG INT we used MRINT
Instead of SSG CON we used MRCON
Reviewer: Single blinded?
Author: Changed as “double-blind”
Reviewer: Line 44: I do not think authors need to spend so much time discussing the SSG in the introduction as it is not really your intervention.
Author: The related sentence was removed.
“ A game-based training approach ensures ecological conditions while providing well-founded holistic development of soccer players.”
Reviewer: Authors have cited a lot of research looking at the acute explosive effects of CHOmr (e.g. sprinting). What is the rationale for why CHOmr would help over the course of an entire game when the effect is lost so quickly?
Author: Detailed explained into the manuscript like below:
“A literature review indicated that rinsing the oral cavity with CHO elicits a distinct metabolic response compared to CHO ingestion [14]. Furthermore, another recent systematic review stated that MR with CHO improves physical performance through psychophysiological effects, directly linking better task-specific activity to activation of the primary sensorimotor cortex due to oral CHO transmission [1]. Therefore, clarifying the specific effects of MR in various SSGs formats may significantly improve players’ performance during training. As noted by Rollo et al. [3], while MR demonstrates efficacy in enhancing performance during intermittent exercises, its influence on soccer-specific formats, namely INT and CON SSGs, requires further investigation, thus highlighting the pioneering nature of this investigation. Finally, current studies show that more studies are needed on the effect of the MR mechanism on soccer performance.”
Reviewer: Authors should remove discussion about energy and metabolism in the context of CHOs for this, as CHOmr is a different neural pathway which authors have highlighted.
Author: We are totally agree with you, the sentence was removed.
“CHOs supply energy to muscles and the central nervous system, preventing fatigue and maintaining physical fitness and technical skills during high-intensity, prolonged exercises, such as soccer matches.”
Reviewer: Did you get informed assent from the parents? Consenting a minor on their own is not customary.
Author: Thank you very much for your patient. Added.
“All the participants and their parents provided written informed consent for participation.”
Reviewer: Figure 1- You cannot read the text, even at 150% magnification. I suggest using abbreviations in the figure so the text can be larger. Then, you can spell out the abbreviations in the caption. The figure looks quite nice though!
Author: Thank you very much reconfigured. The recommended revisions regarding the figures were realized on the figures.
Reviewer: “or crossover study design?”. What is crossover?
Author: In a crossover study design, two interventions (e.g., exercise, methods, and training) are provided to subjects at different time periods, and the sequence of intervention is randomized for each subject. Similarly, the research design of the current study includes two different groups and two different interventions.
Reviewer: Line 119: How are they identically flavored? Was aspartame included in the maltodextrin solution as well?
Author: Both formulations were designed to possess similar flavor profiles to minimize organoleptic differences between the solutions. However, no aspartame or comparable artificial sweeteners were incorporated, and a maltodextrin solution was prepared to render its inherent flavor indistinguishable from that of the placebo solution.
Reviewer: “A single-blinded design was used in this study. Except for the supervisor investigator, the other researchers were blinded to prepare a 6.4% maltodextrin (Protein Ocean, Türkiye) solution (CHOMR) or an identically flavoured aspartame solution (PLA).” Were researchers blinded and not participants? This seems double-blinded based on this statement.
Author: Thank you, we changed.
“In the present study, a randomized, double-blinded, placebo-controlled, or crossover study design was used to assess the effects of MR intervention on psychophysiological responses and kinematic profiles 4-a-side SSGs with MRINT or MRCON formats in adolescent soccer players.”
Reviewer: Line 187: making a figure showing the structure of the games would be helpful
Author: Thank you very much for your suggestion, we added a figure.
Reviewer: What post-hoc test was used?
Author: We do not use any post-hoc test. Instead of post-hoc test, we used effect size due to we have two groups.
Reviewer: Figure 2. These figure look fantastic. I might suggest making the large mean bars more transparent so you can see the individual lines better.
Author: Thank you very much. Changed. The recommended revisions regarding the figures were realized on the figures.
Reviewer: I think an argument author should make in the discussion is that CHOmr may be more beneficial in a fatigued state.
Author: Thank you very much again. That’s fantastic insight. We added.
“Moreover, MR with CHO does not directly supply energy. However, it can influence the central nervous system to signal muscles to work effectively for extended periods, thereby delaying the perception of physical fatigue and maintaining muscle productivity [19]. In conclusion, CON game, as opposed to INT game, requires a configuration physical and mental planning that demands sustained effort for prolonged endurance, so that central fatigue-regulating systems can operate more actively.”
Reviewer: Line 283: authors only saw moderate effects in sprint number and distance. The rest were small. Please remove these from the discussion as a “moderate” effect if the effect size is not >0.5.
Author: Thank you very much. We especially focused on kinematic responses such as total distance (0.91), number of sprints (0.52) and relative metabolic load (0.83). We have already use the effect size >0.5.
Reviewer: Line 294: CHO ingestion should not be discussed in this manuscript as they did not ingest them. Glycogen is not within the mechanism for how CHOmr works.
Author: We are totally agree with you, the sentence was excluded.
“CHO methods include the MR technique [15], which involves briefly swishing a CHO solution in the mouth before spitting it out without ingestion to enhance soccer performance”
“However, some studies have not demonstrated enhancements in sprint performance during team games with CHO ingestion”
Reviewer: Author should spend more time explaining why the limitations matter.
Author: Reorganized our limitations.
“Limitations
The authors fully know the limitations and various variables that may have influenced the analysis results. The current study only assessed young, well-trained male soccer players still developing biological and psychomotor capabilities, limiting its applicability to competitive soccer players from different divisions to varying fitness levels, female soccer players, or older players. Although this study only used a 4-a-side SSG format, researchers could investigate whether the game influences the performance of MR with CHO using several variables, such as the number of players, rules, court dimensions and coach encouragement, which are known to influence SSGs performance. Finally, environmental temperatures can significantly affect physiological and psychophysiological responses during exercise, as the players are trained in hot conditions. Moreover, this study measured HR, enjoyment, and RPE responses, but could also consider psychophysiological variables such as blood lactate level, motivation, cognition, and anxiety, which MR might impact and affect performance. Accepting this limitation and recommending future research to investigate the effects of CHO with MR in a significantly broader target demographic group would increase our understanding of the ergogenic effects of MR. In addition, it would be worth using MR to examine the players’ influence this nutritional strategy has on breaking the fatigue barrier and shifting the psychomotor fatigue threshold towards higher physical loads, at which the fastest reaction, anticipation, and decision-making occur.”
Reviewer: Why discuss outcomes that were not measured in the conclusion?
Author: Reorganized our conclusion.
“Conclusions
Our results support that regardless of the game's structure, the MR intervention can be used as an effective ergogenic supplement for acute game performance enhancement in young soccer players. The CON game elicited higher psychophysiological responses, including perceived exertion, internal training load, and mental effort, whereas the INT game elicited greater enjoyment. Although the MR intervention similarly enhanced performance in the 4-a-side INT and CON games, no significant differences were observed between the groups. Clarifying these responses could offer a complete understanding of the influence of MR on soccer performance. However, this study focused on the acute responses to MR during the micro-cycle. From a practical perspective, future research should explore the different dose effects of regular MR use throughout the training adaptations and recovery. “
